# Prediction of Two-Year Recurrence-Free Survival in Operable NSCLC Patients Using Radiomic Features from Intra- and Size-Variant Peri-Tumoral Regions on Chest CT Images

**DOI:** 10.3390/diagnostics12061313

**Published:** 2022-05-25

**Authors:** Soomin Lee, Julip Jung, Helen Hong, Bong-Seog Kim

**Affiliations:** 1Department of Software Convergence, Seoul Women’s University, 621 Hwarang-ro, Nowon-gu, Seoul 01797, Korea; soominlee@swu.ac.kr (S.L.); jjulip@swu.ac.kr (J.J.); 2R&D Center, Boryung Pharmaceutical Ltd., 136 Changgyeonggung-ro, Jongno-gu, Seoul 03127, Korea; seog9270@boryung.co.kr

**Keywords:** chest CT, radiomic, prognosis prediction, intratumoral, peritumoral, recurrence-free survival

## Abstract

To predict the two-year recurrence-free survival of patients with non-small cell lung cancer (NSCLC), we propose a prediction model using radiomic features of the inner and outer regions of the tumor. The intratumoral region and the peritumoral regions from the boundary to 3 cm were used to extract the radiomic features based on the intensity, texture, and shape features. Feature selection was performed to identify significant radiomic features to predict two-year recurrence-free survival, and patient classification was performed into recurrence and non-recurrence groups using SVM and random forest classifiers. The probability of two-year recurrence-free survival was estimated with the Kaplan–Meier curve. In the experiment, CT images of 217 non-small-cell lung cancer patients at stages I-IIIA who underwent surgical resection at the Veterans Health Service Medical Center (VHSMC) were used. Regarding the classification performance on whole tumors, the combined radiomic features for intratumoral and peritumoral regions of 6 mm and 9 mm showed improved performance (AUC 0.66, 0.66) compared to T stage and N stage (AUC 0.60), intratumoral (AUC 0.64) and peritumoral 6 mm and 9 mm classifiers (AUC 0.59, 0.62). In the assessment of the classification performance according to the tumor size, combined regions of 21 mm and 3 mm were significant when predicting outcomes compared to other regions of tumors under 3 cm (AUC 0.70) and 3 cm~5 cm (AUC 0.75), respectively. For tumors larger than 5 cm, the combined 3 mm region was significant in predictions compared to the other features (AUC 0.71). Through this experiment, it was confirmed that peritumoral and combined regions showed higher performance than the intratumoral region for tumors less than 5 cm in size and that intratumoral and combined regions showed more stable performance than the peritumoral region in tumors larger than 5 cm.

## 1. Introduction

Lung cancer is the leading cause of cancer-related death worldwide [1]. NSCLC is the predominantly diagnosed type of lung cancer, accounting for about 80~85% of all types [2], and the major histological subtypes of NSCLC are adenocarcinomas, squamous cell carcinomas, and large cell carcinomas [3]. In general, NSCLC spreads slowly and gradually throughout the body through the surrounding lymph nodes, and the extent of the spread of NSCLC is described according to the TNM stage considering three factors: the size and location of the tumor (T), the degree of lymph node invasion (N), and the presence of metastasis to other organs (M) [4,5]. Overall tumor stages are defined as stages I to IV by combining the three factors of the TNM stage, and patients with stage I–II NSCLC and some patients with resectable stage IIIA NSCLC undergo curative resection surgery as a treatment [6,7]. However, the recurrence rate after surgery is quite high at 50% [7,8,9]. Therefore, predicting patient outcomes after surgery is important to improve the prognosis of surgery by applying appropriate follow-up management strategies to patients.

Chest CT images are used to diagnose lung tumors, and the appearance of a lung tumor, such as the size and location and the degree of invasion into normal organs, is used as a reference to estimate the risk of the recurrence of lung tumors. Some tumors have typical appearances associated with recurrence and non-recurrence, and these tumors tend to be visually easy to distinguish as to whether or not they are actual recurrences. The non-recurrent tumors are small in size, completely located in the lung parenchyma, and there is no evidence of invasion into normal organs. The recurrent tumors are large, attached to the chest wall or mediastinum, and appear to invade normal organs. However, as shown in Figure 1, non-recurrent tumors and recurrent tumors have similar appearances more frequently in practice. For this reason, the prognosis is difficult when using CT images during visual evaluations. Thus, radiomics, which quantifies the tumor phenotype and visually indistinguishable features, can help detect potentially recurrent tumors through non-invasive biomarkers [10,11].

Several studies have used radiomic features to predict the prognosis of lung cancer patients using CT images, and these studies can be categorized into those that investigate only the intratumoral region or both the intratumoral and peritumoral regions [12,13,14,15,16,17,18,19,20]. In a radiomics study using intratumoral region information, Coroller et al. [12] predicted distant metastasis (DM) of lung adenocarcinoma patients at stages II–III who were treated with radiation. In their study, 635 radiomic features were used, including those based on the intensity, texture, shape, a Laplacian of Gaussian (LoG) filter, and a wavelet filter. Their findings demonstrate that the features of the texture heterogeneity and intensity skewness of the LoG and the wavelets are strongly associated with DM in patients. Coroller et al. [13] predicted the level of the pathological response of patients at stages II-III who underwent neo-adjuvant chemotherapy and radiotherapy before surgery. In that study, 1603 radiomic features were used, including intensity, texture, shape, LoG-filter-based, and wavelet-filter-based features. The study, in this case, showed that the spherical shape of the tumor is important when attempting to predict the pathological response to chemo-radiotherapy. Cong et al. [14] predicted lymph node metastasis in patients at stage IA who underwent curative resection surgery. In this study, 396 radiomic features were used, including intensity, texture, and shape features, indicating that significant features, such as the uniformity of the gray-level co-occurrence matrix and tumor homogeneity, provided predictive information about lymph node metastasis.

In radiomics studies using intratumoral and peritumoral region information, Baek et al. [15] predicted two- and five-year overall survival and disease-specific survival rates for NSCLC patients at stages I-IV who underwent stereotactic radiotherapy. Their study used deep features extracted from the final layer of the encoding path of a U-net trained with CT images and PET-CT images, finding that the encoded features of the U-net depicted interpretable structural and geometric patterns of intratumoral and/or peritumoral structures and that these features could be used to predict survival in cancer patients. Tai et al. [16] predicted distant metastasis in patients at stages Ⅱ–Ⅲ who underwent radiotherapy. The peritumoral rim region was defined as the inner and outer 3 mm tumor margins from the tumor boundary, and the exterior region was defined as a 3 mm to 9 mm outside region. In this study, 2175 radiomic features were used, including intensity, texture, and shape features. Their study found that the peritumoral rim features, which show heterogenetic textures and sharp changes in intensity, can provide significant information for predicting distant metastasis of NSCLC. Wang et al. [17] predicted lymph node metastasis of adenocarcinomas patients at the T1 stage who underwent curative resection surgery and lymph node dissection. The peritumoral regions were defined as 15 mm tumor margins toward the outside from the tumor boundary. In their study, 1946 radiomic features were used, including intensity, shape, and texture features. They found that peritumoral radiomic features are significant when predicting lymph node metastasis in stage T1 lung adenocarcinoma. Khorrami et al. [18] used radiomic features to predict the chemotherapy response of lung adenocarcinoma in patients at stages IIIB–IV. The peritumoral regions were defined as 15 mm tumor margins toward the outside from the tumor boundary. In this study, 1542 radiomic features were used, including intensity, texture, and shape features. Their study found that features reflecting the compactness shape of the tumor and heterogeneous patterns in the intratumoral and the peritumoral regions provide significant information for predicting chemotherapy responses. Vadiya et al. [19] predicted three-year disease-free survival rates of patients at stages I–III who underwent surgery. The peritumoral regions were defined as 15 mm tumor margins toward the outside from the tumor boundary. In this study, 4464 radiomic features were used, including Gabor, Haralick, CoLlAGe, Laws energy, and Laplace features. This study found that the radiomic-based method could identify patients who received the benefit of adjuvant chemotherapy after surgery. Akinci D’Antonoli et al. [20] predicted the recurrence-free survival of stage I–IIB patients who underwent surgery. The peritumoral regions were defined as 20 mm tumor margins toward the outside from the tumor boundary. In this study, 94 radiomic features were used, including Gabor, Laws energy, Laws Laplacian, Haralick, and shape features. It was found that the flatness and irregularity of the shape and the heterogeneity of intratumoral and peritumoral regions were significant predictors of patient survival. These related studies confirmed the ability of the radiomic features of intratumoral and the peritumoral regions to predict prognosis. However, they only investigated specific peritumoral regions up to 2 cm.

In the present study, we propose a method for predicting two-year recurrence-free survival from tumor stages I to IIIA in NSCLC cases using intratumoral and peritumoral radiomic features. To investigate potentially predictive information in the lung parenchyma, we utilize the peritumoral region defined around the tumor. To identify the significant range of the peritumoral region, the peritumoral regions are defined as up to 3 cm at a 3 mm interval. To evaluate the predictive performance according to the tumor size, the results are analyzed by dividing the patient cohort into three size groups.

## 2. Materials and Methods

The pipeline of the proposed prediction model consists of five major steps, as shown in Figure 2. First, the region of interest (ROI) is defined as inside and outside the tumor. Second, radiomic features are extracted from these regions. Third, a set of significant features is identified using the nearest neighborhood analysis (NCA) algorithm. Fourth, patients are classified into recurrence and non-recurrence groups using the support vector machine (SVM) and random forest methods. Finally, Kaplan–Meier curves are used to estimate the probability of recurrence-free survival within two years in predicted patients.

### 2.1. Materials

Preoperative chest CT images of 263 NSCLC patients who underwent curative surgical resection were acquired from the Veterans Health Service Medical Center, Seoul, South Korea. The CT images were obtained from three different CT scanners (SIEMENS SOMATOM Definition AS+, SIEMENS SOMATOM Sensation 64, GE Healthcare LightSpeed ULTRA) using the following scan parameters: 99~129 kVp at 60~534 mAs. Each image had a matrix size of 512 × 512 pixels with in-plane resolutions ranging from 0.54 to 0.83 mm. The slice thickness ranged from 1.0 to 7.5 mm. This study was approved by the Institutional Review Board of the Veterans Health Service Medical Center (IRB File Number: BOHUN 2018-07-009-005). In the acquired data, patient selection was applied, as shown in Figure 3. This study excluded patients without information pertaining to recurrence, those at the T stage and N stage, and patients whose tumors are incorrectly labeled as too large. In addition, according to the opinion of clinicians that it is better to exclude data with large deviations, patients with a tumor larger than 19 cm based on the representative cross-section were also excluded. Finally, the remaining 217 NSCLC patients served as the study group. The patient characteristics are shown in Table 1.

### 2.2. Data Preparation

In the preprocessing step, intensity rescaling and region cropping are conducted on the entire set of CT images, as shown in Figure 4a. The CT images with lung window settings (WW: 1500 HU, WL: −600 HU) are rescaled with gray-scale intensity from 0 to 255 and cropped. In this study, a representative cross-section slice with the longest diameter in the axial view was used. As shown in Figure 4b, the intratumoral region is defined as a segmented region. Lung tumor segmentation was performed semi-automatically by a one-board certified radiologist using in-house software in lung window settings. For tumors that are difficult to distinguish from surrounding structures, a correction was performed on mediastinal window settings (WW: 350 HU, WL: 50 HU). Areas of necrosis and cavities appearing at a low-intensity level within the tumor are included. As shown in Figure 4c, the peritumoral region is defined as the surrounding tumor region from the boundary to 3 cm with a 3 mm interval using the morphological dilation method [21]. The chest wall and mediastinum are excluded by an intensity threshold of −224 HU, which is a value that allows visually distinguishing of the region between the lungs and the chest wall. The primary and secondary bronchi are excluded using airway segmentation [22,23]. In Figure 4d, the combined region is defined as a combination of intratumoral regions and peritumoral regions.

### 2.3. Radiomic Feature Extraction and Selection

Table 2 shows the radiomic features used in this study. From the intratumoral region, 69 radiomic features, including intensity, texture, and shape features, are extracted. In the case of intensity features, it consists of 7 histogram stisticis features and 5 histogram percentile features [24]. The texture feature consists of 14 GLCM features, 22 GLRLM features, and 10 LBP features, and the shape feature has 11 size and roundness-related features [24]. From the peritumoral region, 58 radiomic intensity and texture features are extracted, excluding 11 shape features. In the combined region, 127 radiomic features are used by combining 69 intratumoral and 58 peritumoral radiomic features.

Feature selection is performed using the NCA algorithm, which is based on distance metric learning to find the optimal feature space for classification by defining a distance function [25]. The weight of the feature is determined as the value in each case that minimizes the distance among the intra-class points, and the leave-one-out validation error. Features are incrementally included in the classifier according to their weights, and the final significant feature set is determined as the values that maximize the classification performance based on AUC in the intratumoral region, peritumoral regions up to 3 cm, and combined regions up to 3 cm. In the equivocal case when identifying the maximum classification performance, Youden’s J statistic value is used [26].

### 2.4. Prediction of the 2-Year Recurrence-Free Survival

To predict two-year recurrence-free survival rates, patients are classified into recurrence and non-recurrence groups, and the probability of recurrence-free survival within two years is estimated. First, classification into recurrence and non-recurrence groups is performed using SVM and random forest classifiers. The SVM is a supervised learning method for classification that maximizes the margin between class boundaries and decision boundaries [27]. In this study, the SVM is constructed with the radial basis function (RBF) kernel for nonlinear classification and with the sequential minimal optimization (SMO) algorithm for decision boundary optimization. Random forest refers to an ensemble learning method that uses multiple decision trees for classification [28]. In this study, the random forest is constructed with 60 decision trees trained by bootstrapped samples, and the output of the model is determined by a majority voting technique from the decisions of individual trees. Second, the Kaplan–Meier curve is used to estimate the predicted probability of recurrence-free survival within two years [29]. In addition, comparisons between predicted patient groups are provided as *p*-values calculated through a log-rank test [30].

## 3. Results

### 3.1. Experimental Setting

Here, 217 NSCLC patients of stage I to stage IIIA formed the experimental group, and they were divided into two groups according to recurrence within two years after surgical resection. The stratified five-fold cross-validation method was used to complement the limitations of the small dataset, with each fold constructed by preserving the percentage of data for each class. Matlab 2020b software was used for feature extraction, feature selection, and for recurrence predictions. Kaplan–Meier curves and ROC curves were drawn in Python 3.7 using the scikit-learn library, version 0.23.2.

Classification of patients into recurrence and non-recurrence was performed using the intratumoral, peritumoral, and combined radiomic features, respectively. To identify the ability of radiomic features for prediction, the classification performance of clinical information which is conventionally used for prognosis prediction was compared to that of the radiomic features. In the experiment, the T stage and N stage of the tumor were used as clinical information because the TNM stage represents details of the tumor status, and the M stage was not used because the patients in this study did not have distant metastases. In addition, the analysis of the classification performance according to the tumor size was performed on the radiomic-based classifiers. Classification performance was evaluated by accuracy, sensitivity, specificity, positive predictive value (PPV), and negative predictive value (NPV). True Positive (TP) is an outcome where the model correctly predicts the recurrence, and True Negative (TN) is an outcome where the model correctly predicts the non-recurrence. The area under the receiver operating characteristic curve (AUC ROC) was used to evaluate the classification performance [31].

### 3.2. Classification into Recurrence and Non-Recurrence Group Using Radiomic Features

The classification performance outcomes of radiomic features for two-year recurrence-free survival are shown in Table 3. The performance of each classifier means the average performance of 5-folds, and the performance of each fold was selected from the model with higher performance between the SVM and random forest. In all classifiers, the SVM was selected more than random forest, and SVMs were selected at a rate of 80% in both T stage and N stage and intratumoral radiomic features. For peritumoral and combined radiomic features, SVMs were selected at rates of 74% and 58%, respectively. The T stage and N stage showed lower performance than most radiomic-based classifiers with an AUC of 0.60. Regarding the peritumoral radiomic classifiers, the radiomic features of the 3 mm and 12 mm regions showed significantly better performance than other peritumoral regions with AUC values of 0.66 and 0.63. For the combined radiomic classifiers, the radiomic features of the 6 mm and 9 mm regions showed significantly better performance than other combined regions with AUC outcomes of 0.66 and 0.66. The ROC curves of each classifier are shown in Figure 5. In the curves for each fold, the C fold had a significantly higher AUC for radiomic-based classifiers, and fold A showed a lower AUC for all classifiers. Most of the non-recurrence tumors in the C fold were small and had few vessels in the peritumoral region, but some tumors were at the N1 stage and N2 stage, known as high-risk factors for recurrence. The recurrence tumors in the C fold were large in size and showed various vessel distributions in the peritumoral region, but some tumors were small and at the N0 stage. Therefore, the classification results of the T stage and N stage in the C fold showed that non-recurrent tumors in the N1 stage and N2 stage and recurrent tumors in the N0 stage were misclassified. However, most of these misclassification cases with radiomic features were improved, with these results indicating that radiomic features can represent recurrent and non-recurrent tumors which are difficult to express using the T stage and N stage. In contrast, for the A fold, non-recurrent tumors were large, and some of the tumors had various distributions of vessels in the peritumoral region. The recurrent tumors in the A fold had various sizes and fewer vessels in the peritumoral region, but some of these tumors were at the N0 stage, known to be a low-risk factor for recurrence. Therefore, tumors in the A fold appear to be difficult to classify. To assess the discriminative ability of the radiomic-based classifiers through statistical comparison, the DeLong test was applied. As shown in Figure 5, AUCs for almost all the radiomic-based classifiers are around 0.6. However, in the C fold, the DeLong test found that the AUCs of peritumoral radiomic features 3 mm (*p*-value = 0.0355), 12 mm (*p*-value = 0.0350) and combined radiomic features of 6 mm (*p*-value = 0.0150), 9 mm (*p*-value = 0.0103) were statistically significantly higher than that of the T stage and N stage with *p*-values of less than 0.05.

### 3.3. Probability Estimation of 2-Year Recurrence-Free Survival

The Kaplan–Meier curve in Figure 6 shows the probability of two-year recurrence-free survival at a certain time. The curve in Figure 6a is a real curve drawn with the recurrence information from 217 NSCLC patients. In Figure 6b–f, the curves of the radiomic-based classifiers were better separated than those of the T stage and N stage, and the *p*-values of radiomic-based classifiers were also lower than those of the T stage and N stage. Among the radiomic-based classifiers, it was shown that the curves of the intratumoral and peritumoral 3 mm classifiers were better separated than the curves of the other classifiers. The curves of all classifiers were found to be statistically significant, with *p*-values of less than 0.05.

### 3.4. Classification Performance of Radiomic Feature according to the Tumor Size

As the tumor sizes in our data vary widely from 1 cm to 12 cm, the overall performance evaluation is limited with regard to considering various distributions of tumor sizes. Therefore, the classification performance according to the tumor size was analyzed by dividing the patients into three subgroups according to the size criteria of T stage in the 8th edition of the *AJCC Cancer Staging System* [4]: Group 1, tumors less than 3 cm; Group 2, tumors between 3 cm and 5 cm; and Group 3, tumors exceeding 5 cm. Details of the three subgroups are given in Table 4.

The classification performance of Group 1 is shown in Table 5 (a). For small tumors less than 3 cm in size, the peritumoral and combined radiomic classifiers outperformed the intratumoral radiomic classifier. As shown in Figure 7a, as most tumors showed a similar round shape and homogeneous intensity level, the intratumoral region does not appear to contain useful information for distinguishing recurrence from non-recurrence. On the other hand, the peritumoral region appears to be informative due to the various vessel distributions in the micro-environments of the tumors. For the combined radiomic classifiers, the significant ranges appear to be wider than for peritumoral radiomic classifiers because the intratumoral radiomic features performed poorly.

The classification performance of Group 2 is shown in Table 5 (b). In medium-sized tumors between 3 cm and 5 cm, the intratumoral radiomic classifier performed better than in small-sized tumors. Peritumoral and combined radiomic classifiers performed better than the intratumoral radiomic classifier. As shown in Figure 7b, the intratumoral region and the peritumoral regions appear to be significant for distinguishing between recurrence and non-recurrence, as these regions show various appearances. For combined radiomic classifiers, the significant ranges are considered to be narrower than that of the peritumoral radiomic classifier because intratumoral radiomic features show improved performance.

The classification performance of Group 3 is shown in Table 5 (c). For tumors exceeding 5 cm, the intratumoral radiomic classifiers showed most stable performance outcomes than the peritumoral radiomic classifiers, and the combined 3 mm radiomic classifier showed the highest performance with an AUC value of 0.71. As shown in Figure 7c, for some tumors filling most of the lung parenchyma, it appears to be difficult to use the peritumoral region. For this reason, it is considered that the performance of the peritumoral radiomic classifiers was low and that the significant range of the combined radiomic classifier was narrow at 3 mm.

## 4. Discussion and Conclusions

This study proposed a prediction model for predicting the two-year recurrence-free survival of patients with NSCLC using radiomic features of intratumoral and peritumoral regions. We used radiomic features to find that the peritumoral regions have potential predictive ability for predicting two-year recurrence-free survival. This study identified that combining the intratumoral radiomic features and peritumoral radiomic features improved the predictive performance, and the combined regions of 6 mm and 9 mm showed the highest performance. As a result of a performance analysis according to the size of the tumor, it was found that radiomic features in a significant range differed according to the size of the tumor. For tumors less than 5 cm in size, it was confirmed that the peritumoral and combined radiomic features outperformed the intratumoral radiomic features. For tumors larger than 5 cm, we found that the intratumoral radiomic features showed the most stable performance outcomes compared to the peritumoral radiomic features.

The significant radiomic features of the intratumoral and peritumoral regions are summarized in Table 6. In the intratumoral region, texture features accounted for most of the significant features, and low-intensity features and shape features for size and roundness were also significant. In CT texture analysis studies, it is known that intensity and texture features reflect tumor heterogeneity, as determined by the histopathological phenotype and genotype according to studies by Bashir et al. and Lubner et al. [32,33]. Therefore, it is considered that the texture features reflecting the heterogeneity of the tumor and the intensity features reflecting the existence of necrosis were mainly selected as significant features for the predictions here. In the peritumoral region, the intensity features accounted for most of the significant features, and the texture features were also significant. As the heterogeneity of the peritumoral region reflects evidence of tumor spread or invasion in the microenvironment, the intensity, and texture features quantifying the heterogeneity have been shown to be significant [34].

In the results of this study, we found that some specific folds resulted in inferior performance compared to others for all classifiers. For non-recurrent tumors, there was a tendency to misclassify tumors that were relatively large, irregular in shape, and showed various vessel distributions around the tumor as recurrent tumors, whereas for recurrent tumors, relatively small tumors located in the lung parenchyma and the distribution of small vessels around the tumor tended to cause them to be misclassified as non-recurrent tumors.

In prognosis prediction studies of NSCLC, the peritumoral regions were investigated in an effort to improve the prediction performance. The peritumoral regions investigated in the previous studies were limited up to 2 cm outside of the tumor, with those studies using a specific peritumoral region. Tai et al. used the peritumoral region, defining inner and outer 3 mm regions based on the tumor boundary as a “rim” region and the 3 mm to 9 mm peritumoral region as an “exterior” region [16]. Wang et al., Khorrami et al., and Vadiya et al. defined the peritumoral region as 15 mm toward the outside and Akinci D’Antonoli et al. defined the peritumoral region as 2 cm toward the outside [17,18,19,20]. These studies did not investigate about the significance of the peritumoral region according to tumor size. In contrast, our study investigated the peritumoral region from the boundary to 3 cm with a 3 mm interval, and we found that peritumoral regions exceeding 2 cm were significant for tumors of a specific size. For tumors less than 3 cm in size, the combined regions of 21 mm and 24 mm were significant, while for tumors between 3 cm and 5 cm, the peritumoral region of 27 mm was significant.

The size of the dataset in this study is limited in terms of its ability to reflect the various characteristics of tumors. Therefore, in future work, we will use an external validation set to validate the proposed method externally. This future study will also take into account the clinical features of the tumors or patients to improve the performance outcomes. In CT radiomics studies, it is known that there is an effect depending on the variability of imaging acquisition parameters and post-process variables; however, this study did not analyze them. This analysis will need to be considered in future work.

## Figures and Tables

**Figure 1 diagnostics-12-01313-f001:**
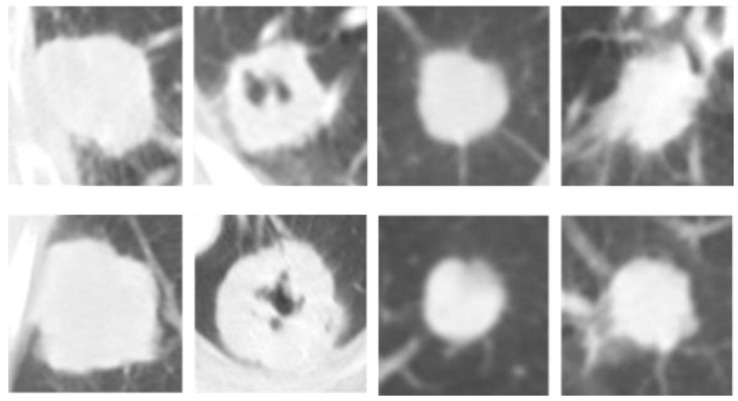
Examples of lung tumors with similar appearance between recurrence and non-recurrence. Non-recurrence tumors are in the first row, and recurrence tumors are in the second row.

**Figure 2 diagnostics-12-01313-f002:**
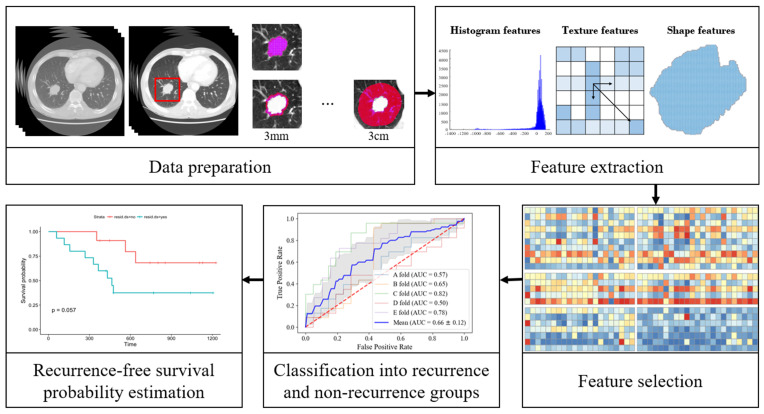
The pipeline of radiomic-based prediction model for 2-year recurrence-free survival prediction.

**Figure 3 diagnostics-12-01313-f003:**
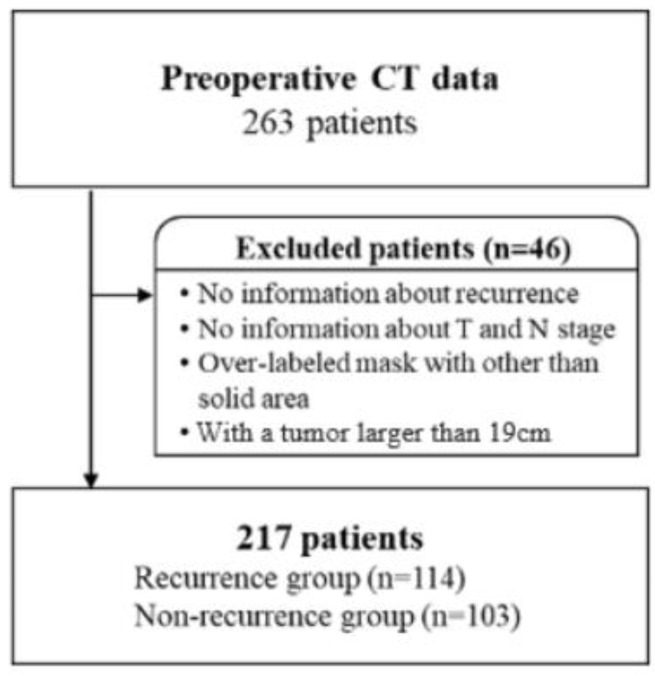
The criteria of patient selection.

**Figure 4 diagnostics-12-01313-f004:**
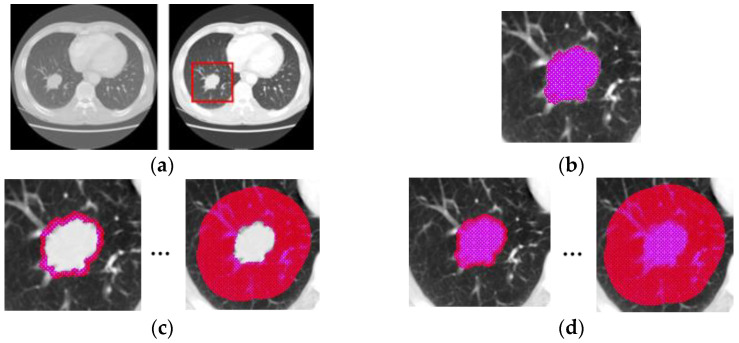
Illustration of data preparation. (**a**) Preprocessing and (**b**–**d**) region definitions for CT images.

**Figure 5 diagnostics-12-01313-f005:**
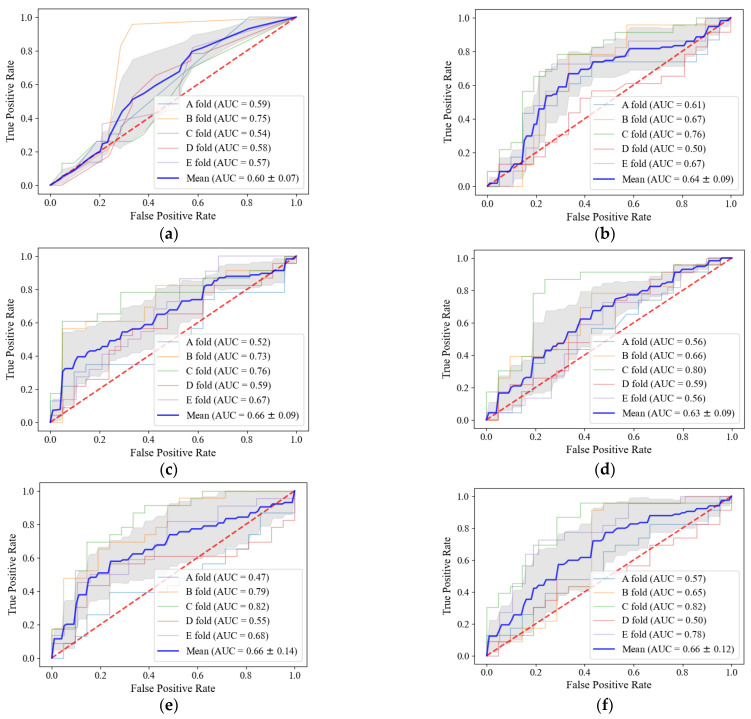
ROC curves for each classifier. The blue line curve indicates the mean value of the 5-fold cross-validation results, and the gray band indicates the AUC variance of the 5-fold cross-validation: (**a**) T and N-stages; (**b**) Intratumoral radiomic classifier; (**c**) Peritumoral 3 mm radiomic classifier; (**d**) Peritumoral 12 mm radiomic classifier; (**e**) Combined 6 mm radiomic classifier; and (**f**) Combined 9 mm radiomic classifiers.

**Figure 6 diagnostics-12-01313-f006:**
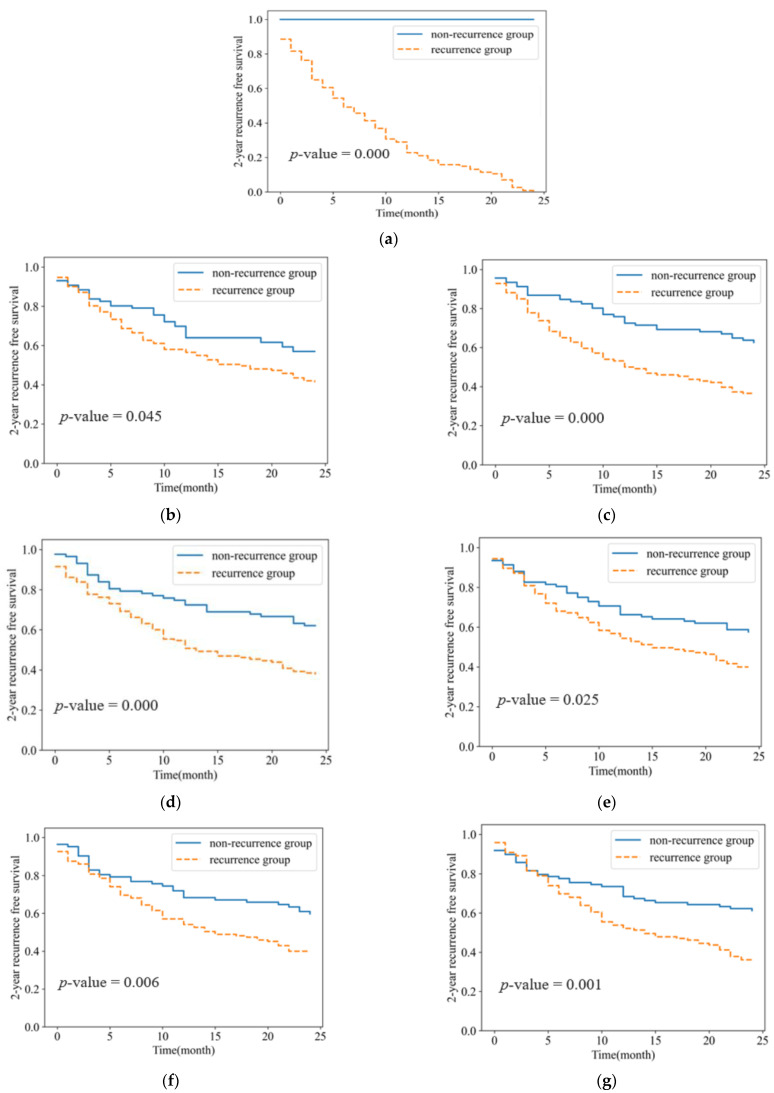
Kaplan–Meier curves for 2-year recurrence-free survival: (**a**) Real curve; (**b**) T stage and N stage; (**c**) Intratumoral radiomic classifier; (**d**) Peritumoral 3 mm radiomic classifier; (**e**) Peritumoral 12 mm radiomic classifier; (**f**) Combined 6 mm radiomic classifier; (**g**) Combined 9 mm radiomic classifier.

**Figure 7 diagnostics-12-01313-f007:**
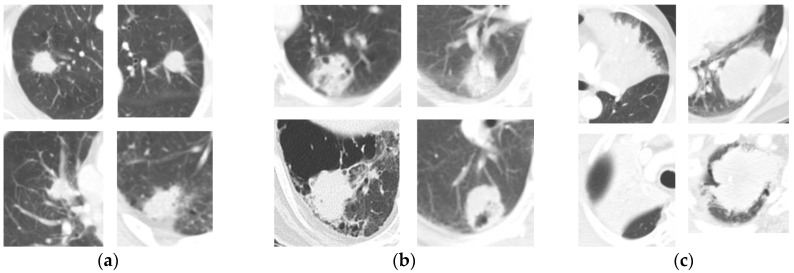
Appearance of lung tumors according to tumor size groups on CT images: (**a**) Group 1; (**b**) Group 2; (**c**) Group 3.

**Table 1 diagnostics-12-01313-t001:** Patient characteristics.

Patient Characteristics		Total (n = 217)	Non-Recurrence (n = 103)	Recurrence (n = 114)
**Age**		73.14 (62–89)	72	74.3
**Gender**				
	Male	212 (98%)	99	113
	Female	5 (2%)	4	1
**Histology**				
	Adenocarcinomas	89 (41%)	47	42
	Squamous cell carcinomas	128 (59%)	56	72
**T stage** ^1^				
	T1	90 (41%)	56	34
	T2	113 (52%)	43	70
	T3	14 (6%)	4	10
**N stage** ^1^				
	N0	122 (56%)	71	51
	N1	56 (26%)	21	35
	N2	39 (18%)	11	28

^1^ The T stage and N stage of tumors were determined by the American Joint Committee on Cancer (AJCC) staging system, 6th and 7th editions.

**Table 2 diagnostics-12-01313-t002:** List of radiomic features used in this study.

Categories	Sub-Categories	Features
Intensity	Histogram Statistics (7)	mean, std, min, max, skewness, kurtosis, entropy
	Histogram Percentile (5)	5%, 25%, 50%, 75%, 95%
Texture	GLCM features (14)	mean and std dev pairs of ASM, contrast, sum average, sum variance, sum entropy, entropy, difference entropy
	GLRLM features (22)	mean and std dev pairs of short and long run emphasis, low and high gray-level emphasis, non-uniformity, run percentage, etc.
	LBP features (10)	local binary patterns using 10 visual descriptors.
Shape	Size and Roundness (11)	area/perimeter ratio, convex area, eccentricity, Euler number, major-minor axis ratio, major axis length, minor axis length, area, perimeter, curvature

**Table 3 diagnostics-12-01313-t003:** Classification performance of radiomic features for 2-year recurrence-free survival.

Classifier		ACC	SEN	SPEC	AUC	Classifier		ACC	SEN	SPEC	AUC
T stage and N stage		58.61	68.73	47.49	0.60	Intratumoral radiomic features		63.23	70.82	55.03	0.64
Peritumoral radiomic features	3 mm	61.32	67.64	54.33	0.66	Combined radiomic features	3 mm	62.68	67.16	57.65	0.65
	6 mm	57.70	65.10	49.59	0.59		6 mm	60.18	68.27	51.23	0.66
	9 mm	58.60	68.04	48.14	0.62		9 mm	62.78	66.64	58.65	0.66
	12 mm	58.60	64.90	51.59	0.63		12 mm	58.05	63.36	51.99	0.64
	15 mm	57.58	59.72	55.04	0.58		15 mm	58.68	64.85	51.92	0.65
	18 mm	58.09	64.94	50.36	0.61		18 mm	60.78	66.63	54.40	0.65
	21 mm	57.75	69.24	44.99	0.60		21 mm	59.67	60.32	58.96	0.64
	24 mm	58.75	67.93	48.52	0.60		24 mm	61.86	66.60	56.68	0.65
	27 mm	58.06	70.98	43.76	0.60		27 mm	62.43	71.48	52.47	0.64
	30 mm	56.49	69.93	41.57	0.60		30 mm	60.57	67.48	53.01	0.64

**Table 4 diagnostics-12-01313-t004:** Details of the patient groups according to the tumor size.

Group	Tumor Size Criteria (cm)	Number of Patients (n = 217) (Recurrence/Non-Recurrence)	Median Tumor Size (cm)
Group 1	<3 cm	88 patients (35/53)	2.22 cm (±0.48)
Group 2	≥3 cm and <5 cm	83 patients (53/30)	3.77 cm (±0.59)
Group 3	≥5 cm	46 patients (26/20)	6.78 cm (±1.8)

**Table 5 diagnostics-12-01313-t005:** Classification performance by tumor size group.

(a)	**Group 1 (tumor size < 3cm)**					
	**Classifier**		**ACC**	**SEN**	**SPEC**	**AUC**
	Intratumoral radiomic features		53.70	39.92	59.25	0.47
	Peritumoral radiomic features	3 mm	58.81	52.58	65.24	0.61
		12 mm	61.24	58.11	66.62	0.67
	Combined radiomic features	21 mm	66.2	53.94	80.78	0.70
		24 mm	62.02	57.12	69.98	0.67
(b)	**Group 2 (3 cm ≤ tumor size < 5 cm)**					
	**Classifier**		**ACC**	**SEN**	**SPEC**	**AUC**
	Intratumoral radiomic features		75.96	85.13	49.29	0.68
	Peritumoral radiomic features	18 mm	66.44	73.59	55.95	0.70
		27 mm	71.48	80.51	58.81	0.73
	Combined radiomic features	3 mm	70.43	83.14	47.81	0.75
		6 mm	65.11	76.88	44.43	0.72
(c)	**Group 3 (5 cm ≤ tumor size)**					
	**Classifier**		**ACC**	**SEN**	**SPEC**	**AUC**
	Intratumoral radiomic features		63.40	82.67	42.50	0.63
	Peritumoral radiomic features	3 mm	61.86	82.00	32.50	0.66
		24 mm	61.45	86.83	27.50	0.55
	Combined radiomic features	3 mm	61.73	83.67	42.50	0.71
		6 mm	51.30	84.33	15.00	0.64

**Table 6 diagnostics-12-01313-t006:** List of significant radiomic features of intratumoral and peritumoral regions.

**Intratumoral Radiomic Features**	
**Category (#n)**	**Feature**
Intensity (2)	Histogram 25% Percentile
	Histogram 5% Percentile
Texture (8)	GLCM Sum Variance
	GLRLM Long Run Emphasis
	GLRLM Long Run High Gray-level Emphasis
	GLRLM Long Run Low Gray-level Emphasis
	GLRLM Low Gray-level Emphasis (std)
	GLRLM Short Run Emphasis (std)
	GLRLM Long Run High Gray-level Emphasis (std)
	LBP #08
Shape (3)	Major Axis Length
	Major-minor Axis Length Ratio
	Convex Area
**Peritumoral Radiomic Features**	
**Category (#n)**	**Feature**
Intensity (5)	Histogram 75% Percentile
	Histogram 95% Percentile
	Histogram Std
	Histogram Min
	Histogram Mean
Texture (4)	GLCM ASM (std)
	GLRLM Long Run High Gray-level Emphasis
	GLRLM Run Percentage (std)
	GLRLM Short Run Low Gray-level Emphasis (std)

#n means the number of features belongs to each category.

## Data Availability

Not applicable.

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
