# Peer review of "Prediction of Two-Year Recurrence-Free Survival in Operable NSCLC Patients Using Radiomic Features from Intra- and Size-Variant Peri-Tumoral Regions on Chest CT Images"

_diagnostics, 2022, doi:10.3390/diagnostics12061313_

Round 1
Reviewer 1 Report
This paper presents a semi-automated method for the semi-automated analysis of NSCLC intra and peri-tumour areas from CTs with the aim of predicting survival.
The paper is clear and well written though there are two severe problems that make me seriously doubt about the scientific soundness of the method (see point A and B).
Plus, there are some minor point (see minor issues). I would suggest carefully revising the work, with the aid of some machine learning expert.
- A) Besides the missing infos, there is a severe problem, generally called "information leakage", that makes me doubt about the obtained performance.
Indeed, while authors properly describe the way they define the positive patients (recurrence within two years) and apply stratified 5-fold cross validation to have an unbiased evaluation,
the crucial problem is that feature selection is performed a priori on the whole dataset, which includes both training and test samples.
In this way, there is information leakage from the test set and the performance are obviously inflated.
To explain better, the algol-like procedure should be:
- P = positive samples, N = negative samples
- labels(x) = function that returns the true labels of the samples in the sample set x
- f_i = p_i U n_i [for i = 1,.., 5] i-th stratified fold, composed of a subset of positive patients (p_i) and a subset of negative patients
pred = empty # all the predictions will be saved here
GT = empty # all the true labels will be saved here
for each i in 1:5: # for each fold
test_fold = p_i U n_i # define the test fold
# extract the true labels for the current test fold and add it to those already stored for preceding folds
GT = GT U labels(test_fold)
train_fold = p_{j \neq i} U n_{j \neq i} # define the training fold
# find the best features by NCA ON THE TRAINING SET!!!
new_features = NCA(train_fold)
# reformat the training and test set to keep only seleted features
selected_train = train_fold(, new_features)
selected_test = test_fold(, new_features)
# train the classifier
Classifier = fit_classifier(selected_train)
# test on the current fold and add predictions to those already computed
pred = pred U predict(classifier, selected_test)
endfor
# finally compute performance measures
sensitivity(GT,pred)
specificity(GT,pred)
AUROC(GT,pred)
AUPRC(GT,pred)
PPV(GT,pred)
NPV(GT,pred)
- B) no code is provided which, again, make me doubt about the model structure and the way the procedures are implemented.
The code is written with MATLAB, which is very powerful but sometimes tricky.
As an example, authors train RFs with 60 trees; WHY did they choose this value? I suppose they opted for the hyperparameter optimization option, which is provided by MATLAB.
But, again, this should be done for each training fold and, in practice, it is impossible that the same number of trees is chosen for different folds.
Minor issues:
- Line 12: the acronymn NSCLC should be defined somewhere; I suggest doing it in the abstract (Line 12), where authors introduce non-small cell lung can- 12
cer
- Section Material, Line 148-150: authors exclude "patients whose tumors are incorrectly labeled as too large". How can it be? please explain.
Further, authors say: "In addition, patients with a 19cm tumor based on the representative cross-section were also excluded".
Why do the author use dthis threshold? And why do they remove these patients? Any choice in the cohort selection should be motivated
- Section Data preparation, line 63: "the intratumoral region is defined as a segmented region".
The segmentation method seems an ad hoc method. I would add a sentence at the beginning of the section specifying that the data preparation involves a semi-automated processing,
where the expert is aided by automated image-enhancement procedures.
- Section Methods: this is the problematic section as explained above

Reviewer 2 Report
The study proposes Intra- and Size-Variant Peri-tumoral Regions to predict prognosis of NSCLC Patients. Subgroup analyses based on tumor size are performed. This idea is interesting, but the content and design of the article still need to be further improved:
The description of method is not specific:
- Information such as scan parameters were not provided
- How to do ROI segmentation, manual or semi-automatic?
- Is the feature extraction based on IBSI standards?
- What features are included in each classifier?
- What method is each classifier based on? svm or rf
The experimental design can be further improved:
- No external validation
- No feature consistency analysis
- Only 2D features are considered
- It may be considered to extract features directly from the entire combined region
- Statistical p-values need to be provided when comparing the predictive performance between different classifiers
Reviewer 3 Report
1. The study presented is retrospective and comes from a single institute with a limited number of patients. As such, the proposed radiomics model may be of limited use. Prospective studies should be considered, and such studies would be very feasible and will almost certainly result in far more viable and clinically relevant results.
2. In terms of the studied cohort, additional patient characteristics, e.g., co-morbidities, smoking status, treatment modality, amongst many other risk factors should be disclosed. Only in this way can selection bias critics be silenced.
3. As the tumors under investigation had such a wide variety of sizes, caution should be used when applying radiomics analysis to those with small volumes given the well-known fact that radiomics lacks relevance and may even be misleading for small size volumes.
4. More details on how the radiomics features were extracted and computed need to be provided, considering it is well known that there exists a wide array of variables within this process, such as quantization algorithms and normalization schemes among a great number of others. Only so doing can the study be reproduced.
5. The statistical analysis that the authors took is problematic. Given that the majority of the statistical difference tests were conducted within a setting where multiple comparisons arise, p-values for significance should be adjusted accordingly, which may, in turn, undermine some of the results achieved in this study.
6. As reported in the manuscript, AUC for almost all the developed classifiers is somewhere around 0.6, which might be, strictly speaking, superior to random guessing, but really not much better. The authors are recommended to apply the DeLong test of the AUCs to assess the discriminative ability of the developed classifiers so that definite conclusions if any could be drawn.
7. A number of previous studies have shown the impacts on CT radiomics analysis due to variability of imaging acquisition parameters and post-process variables (PMID: 28112418; PMID: 24772210; PMID: 33598943). Exploring how these factors affect the generalization of the claimed findings in detail may be out of the scope of the current study, but a discussion of this is definitely warranted in a revised version of this manuscript.
8. There are typos/grammatical slips throughout the paper. The manuscript could be further improved by asking senior professionals to revise language for style and grammar problems.
Reviewer 4 Report
The manuscript “prediction of two-year recurrence-free survival in operable NSCLC patients using radiomic features from intra- and size- variant peritumoral regions on chest CT images” reports an interesting study on estimation of probability of two-year recurrence-free survival using CT images of 217 NSCL lung cancer patients at stages I-IIIA who underwent surgical resection. This study proposes a prediction model using radiomic features of the inner and outer regions of the tumor.
- Figure 1 is not clear, and the legend is confusing. Authors should produce clear images and should indicate the tumor area and size in each image.
- Line 51-55, it is not quite obvious through images to distinguish recurrence and non-recurrence tumors.
- Line 67-69, needs citation from studies as mentioned.
- Line 330-332, this statement needs strong citations.
- Line 353-357, authors should elaborate the ‘significance’ of peritumoral region based size.
Reviewer 5 Report
Overall the article is well written and address an interesting topic.
However, I have a few suggestions:
- the "p" in "p-value" must be written in italics lower-case letters throughout the text, as well as in the Figures.
- Introduction, line 39: please add reference for NSCLC subtypes.
- Introduction, line 93: replace "distance" with "distant".
- Introduction, line 94-96: please re-state this sentence, as it is not clear the region differentiation; also, replace "boundary" with "margins".
- Introduction, line 101, 111, and 117: please re-state the sentence, as it is not clear.
- Introduction line 115, and Discussion line 352: the correct Author surname is Akinci D'Antonoli; please change.
- Materials, line 148: please re-state the sentence "those at T and N stages", as it is not clear what the Authors would like to say.
- Materials: please change "patients with a 19cm tumor" to "patients with a tumor larger than...", and please specify the lesion range, mean lesion diameter and maximum lesion dimension cut-off; please reflect this change also in Figure 3. Also, explain why large tumors were excluded.
- Results, line 229: "to evaluate" what? please state.
- Results, line 274: please replace "Because" with "As".
- Discussion, line 334: please replace "Because" with "As".
- Discussion, line 334-336: please re-state the sentence, as it seems to be missing the conclusion.
- Discussion, line 361: please add "this future study" to the sentence, as it is ambiguous.
- Figure 1: I suggest to remove the "a" column, leaving only the "difficult to differentiate" nodules; please reflect this change also in the text and in Figure caption.
- Table 5: please add spaces between words and symbols in "(3cm≤tumor size<5cm)" and in all others table subheadings.
- Throughout the text, Figures and Tables are cited both as "1(a)" and "1 (a)" (the latter one with a space inserted between number and letter); I suggest to choose the first format "1(a)" (whithout the space in between), and to stick to it in all the manuscript, correcting the mistakes.
- References: must be written all in the same format, according to the Journal requirements.
Round 2
Reviewer 1 Report
Authors addressed all my concerns.
Author Response
Thank you for your comment.
Reviewer 2 Report
- What software was used to extract features and is there a detailed description of these features?
- It is unclear which features are incorporated in the classifier provided by table3 and which classifier is used.
- I would like to know the ICC results of the features obtained by two radiologists.
- I still don't see statistical results for comparisons between different models. The Delong test is recommended for comparison
Reviewer 3 Report
In this revised version of the previously submitted manuscript, the authors merely admitted the issues being identified in the initial review, and no concrete attempts have been made to rectify or address them. This manuscript does not appear to make a major contribution as in its current form.
Author Response
Unfortunately it was not possible to revise the manuscript to reflect all your suggestions.
Reviewer 4 Report
The authors responded reviewers' comments satisfactorily. I recommend this revised version for publication.
Author Response
Thank you for your comment.
Round 3
Reviewer 2 Report
- It is unclear which features are incorporated in the classifier provided by table3. Please add it to the manuscript or supplementary material.
- It is recommended to provide the Delong‘ test results between difference Classifier at three tumor size group (table5).